# Options in Pregnancy to Increase ActiveLy Sitting (OPALS) Feasibility Study

**DOI:** 10.3390/ijerph18115673

**Published:** 2021-05-26

**Authors:** Caterina Fazzi, Fiona C. Denison, David H. Saunders, Jane E. Norman, Rebecca M. Reynolds

**Affiliations:** 1Tommy’s Centre for Maternal and Fetal Health, MRC Centre for Reproductive Health, Queen’s Medical Research Institute, University of Edinburgh, Edinburgh EH16 4TJ, UK; caterina.fazzi@umce.cl (C.F.); Fiona.denison@ed.ac.uk (F.C.D.); 2Department of Physical Education, Sports and Recreation, Metropolitan University of Educational Sciences, Santiago 7760197, Chile; 3Physical Activity for Health Research Centre (PAHRC), Institute for Sport, Physical Education and Health Sciences, University of Edinburgh, Edinburgh EH8 8AQ, UK; dave.saunders@ed.ac.uk; 4Health Sciences Faculty Office, University of Bristol, Bristol BS8 1UD, UK; jane.e.norman@bristol.ac.uk; 5The British Heart Foundation/University of Edinburgh Centre for Cardiovascular Science, University of Edinburgh, Edinburgh EH8 8AQ, UK

**Keywords:** exercise, obesity, pregnancy, sedentary behaviour

## Abstract

Background. A negative association between obesity and pregnancy outcomes has been described, as well as between time sedentary and pregnancy outcomes. Most interventions based on physical activity involving obese pregnant women have failed in improving pregnancy outcomes. Exchanging time spent in sedentary activities with time spent in light-intensity activities, performed in a home-based setting, might help morbidly obese pregnant women. We aimed to assess the feasibility of an exercise intervention. Methods. An exercise intervention for morbidly obese pregnant women was designed involving morbidly obese pregnant women. Pregnant women with BMI ≥ 40 kg/m² with 20 or less weeks of gestation were invited to take part in the OPALS Feasibility Study. A home-based approach was employed. Participants were asked to perform the intervention for at least 12 weeks, and to register their performance in an activity diary. After the intervention, participants were asked to return the activity diary and answer a feasibility questionnaire. Results. In the intervention, 28 participants took part. Six women completed the intervention for 12 weeks or more. All declared they intended to keep on doing the intervention. All women reported that the exercises made them feel better. Conclusion. Empowering, and involving morbidly obese pregnant women in taking care of themselves and giving them realistic tasks to do on their own and around their environment helps to increase commitment, as does avoiding the effect of their own weight whilst exercising. A 20% of compliance was observed in this study, which might be explained by the difficulties that pregnancy and excess weight mean. Thus, for future studies, we suggest adding a supervision plan to increase that number.

## 1. Introduction

Obesity is increasing around the world, including among pregnant women [1,2,3,4]. A negative association between obesity and pregnancy outcomes for mothers and infants has been observed [5,6,7,8,9], and the risks are greater with a higher body mass index (BMI) [2,10]. Additionally, a negative association has also been reported between time spent sedentary and pregnancy outcomes [11]. Effective strategies to help obese pregnant women are required to reduce the risks and the consequences of obesity and sedentary behaviour during pregnancy.

Studies have observed that obese women have low adherence to exercising during pregnancy [12,13,14], taking into account that it may be more challenging for morbidly obese women (BMI ≥ 40 Kg/m²), compared to lean women. However, some evidence shows that home-based physical activity interventions are associated with increasing compliance. Exercising in a familiar and comfortable setting is of particular interest for obese pregnant women [15]. Modest attention has been set on light-intensity activities and sedentary behaviour during pregnancy [16], even though promoting pregnant women to spend more time in light-intensity activities would directly reduce the time they spend sedentary, encouraging maternal and child health benefits [11,17].

Involving participants in the design, development, and the assessment of an exercise intervention, getting feedback right from the protagonists, might be a helpful approach to improve feasibility [18].

Based on the literature, we observed a lack of available interventions in pregnancy focused on reducing sedentary time [11]. Contrarily, different interventions aiming to increase physical activity among overweight or obese pregnant women were found, nevertheless most have failed in improving pregnancy outcomes [12,13]. We believe that a feasibility study is necessary to calculate essential factors, such as recruitment rate, adherence, and compliance, to inform the design of a randomised controlled trial [19].

We aimed to assess the feasibility of an active sitting exercise intervention for morbidly obese pregnant women, which was designed using a patient involvement in research method. We hypothesised that using a home-based exercise intervention including simple, safe, and mostly sitting exercises would encourage at least half of the recruited morbidly obese pregnant women to accomplish the whole proposed exercise plan. 

## 2. Materials and Methods

### 2.1. Design

We used a patient involvement in research method to design a realistic, feasible, suitable, interesting, and motivating exercise intervention, specifically for pregnant women with morbid obesity.

After the exercise intervention was designed and tested with morbidly obese pregnant women, we conducted the present study to assess the feasibility of the intervention.

### 2.2. Ethics Approval

We declare that the present study was carried out following the rules of the Declaration of Helsinki of 1975. The OPALS Study received ethical approval from the Research Ethics Committee on 11 September 2017, reference 17/SS/0101, protocol number AC17053, Integrated Research Application System (IRAS) project ID 228472 and Lothian Research and Development (R&D) approval number 2017/0248 on 26 September 2017.

### 2.3. Setting

A sample of 23 women with BMI ≥ 40 and a singleton pregnancy, able to perform physical activity, attending the Tommy’s Antenatal Metabolic Clinic at the Royal Infirmary of Edinburgh between 14 March 2017 and 30 May 2017 took part in the design of the exercise intervention, by informing regarding comfort, intensity based on the Borg’s Scale [20], and number of repetitions considered optimal for each exercise.

Once the final designed exercise plan was completed, and the ethical approval was obtained by the South East Scotland Research Ethics Committee, we started to conduct the OPALS (Options in pregnancy to increase actively sitting) feasibility Study by recruiting participants at the Tommy’s Antenatal Metabolic Clinic, in the Royal Infirmary of Edinburgh, between 10 October 2017 and 5 June 2018.

### 2.4. Characteristics of Participants

In the OPALS Feasibility Study were included pregnant women with BMI over 40 kg/m², attending the Tommy’s Antenatal Metabolic Clinic, during the first trimester of gestation (less than 20 weeks of gestation), between 16 and 50 years old, with a healthy singleton pregnancy, with the ability to provide informed consent.

Women were excluded from the OPALS Feasibility Study if having any contraindication to perform exercise, according to the consultant. 

### 2.5. Description of Intervention

We designed a series of six low-intensity exercises as alternatives to sedentary behaviour, based on FITT (frequency, intensity, time and type of exercise) guidelines [21], involving most of big muscle groups, to work them alternatively, following the training principles [22], which could be done while sitting. We proposed a range of repetitions, time, and sets, which should last between 30 to 40 min, including pauses, acknowledging what is recommended for obese pregnant women [23,24].

During personalised interviews women learned how to perform each exercise, and tested all of them, reporting their feelings regarding intensity, using the 1–10 Borg’s Rating of Perceived Exertion Scale [20], comfort, perception on the number of repetitions, or time holding the exercise, and sets. Women were also asked if they had any comments or suggestions to improve the intervention. All answers were collected in a specially designed Feedback Form.

During the first antenatal appointment, after confirming eligibility, midwives gave the OPALS Study Information Sheet to potential participants, and briefly explained the intervention. The information sheet detailed all aspects of the study. Interested patients were referred to the researcher, who held a personalised interview, where patients were explained all aspects of the exercise intervention in more depth. Those who agreed to take part signed the consent form and received the document wallet, including the participant’s copy of the consent form, the Participant Information Sheet, Exercise Strategy Guideline, the Exercise Strategy Activity Diary, a laminated summary of the exercises with a magnet, and the ball required to perform the exercises. After agreeing to take part, the researcher demonstrated each exercise, and performed all the exercises with each participant, following the Exercise Strategy Guideline. The researcher also explained how to fill the Exercise Strategy Activity Diary.

An information sheet for every recruited woman’s general practitioner (GP) was sent to inform that the patient was taking part in the OPALS Feasibility Study, and briefly what the intervention involved.

The OPALS Exercise Strategy Guideline handed to the participants, provided information regarding the risks associated with obesity and too much time spent in sedentary behaviour during pregnancy, and the potential benefits of the exercise intervention. It also described in detail, and with drawings, each of the six exercises included in the intervention. The OPALS Exercise Strategy Activity Diary was a 32-page booklet, specially designed for the OPALS Study, which included instructions on how to complete the diary, followed by 14 weeks to complete daily, according to the performance, plus a space to write extra comments.

After 12 weeks from the first interview with the researcher, participants were contacted by text message or phone call, asking to bring the Exercise Strategy Diary to the next appointment at the Tommy’s Antenatal Metabolic Clinic. In that appointment the researcher contacted the participants, retrieved the diaries, and administered the OPALS Feasibility Questionnaire.

The OPALS Feasibility Questionnaire was self-administered and designed as part of the study to collect information regarding the feasibility of the intervention. Consisted of 15 questions.

### 2.6. Exercise Intervention

The exercise intervention consisted of six exercises involving most of the big muscle groups. Five exercises were designed to be performed while sitting, and one standing (squats). Four exercises required a medium size (22 cm diameter) soft ball which was provided. Three exercises were mainly for legs, two for arms and one for abdominals. All six exercises were planned to be done in ten repetitions and two sets with a total recovery pause between sets.

Participants were asked to perform the intervention for at least 12 weeks, three or more times per week. Doing the complete intervention took between 45 to 60 min, depending on the number of repetitions or time holding, as most of exercises had suggested ranges. Length of time also depended on the time each participant took to recover between sets and exercises.

### 2.7. Materials

A medium size (22 cm diameter) soft ball (provided), and a steady and sturdy chair.

### 2.8. Analysis

The feasibility results of the exercise intervention were analysed descriptively, considering mostly the Activity Diary data and the Feasibility Questionnaire. The main focus of the analysis was set on quantifying recruitment rate, defined as the proportion of women who consented in taking part in the intervention from all women approached. Adherence was defined as to what extent the participants agreed with the intervention [25], and compliance was defined as at what extent the participants effectively followed what were instructed to do [25]. A qualitative analysis was also conducted considering data described in the comments written by participants in the diary. Specifically, the number and proportion of women completing the intervention, and the number and proportion of women who declared to keep on doing the exercises were described. The number and proportion of participants who quitted the intervention were quantified, along with the reasons to quit. 

### 2.9. Sample Size

As a feasibility study, a formal size calculation was not required, though a pragmatic decision was made for the sample size, mostly based on time schedule. We aimed to recruit as many women as possible, considering that the intervention was designed to last 12 weeks [26].

## 3. Results

The OPALS Feasibility Study was based in the exercise intervention designed previously. After approaching 69 women meeting the eligibility for inclusion, 30 women were recruited (recruitment rate = 43.5%). Of these, two women did not attend their appointment in the Clinic to return the diary and administer the completion questionnaire, leaving a total sample of 28 subjects (Figure 1).

Of all women, 17/28 (60.7%) returned the Exercise Strategy Activity Diary, whilst 11 did not (39.3%). 

As Figure 2 shows, six (21.4%) women completed the intervention for 12 weeks or more. Five (17.8%) women performed the intervention for six to 11 weeks. Three (10.7%) did not start immediately after the first interview, but they started later and had done the exercises for only three to five weeks, when the questionnaire was administered, and the diaries collected. The other 14 (50.0%) women did not perform the intervention for more than two weeks, of these, one said she would keep on doing it (Table 1).

Analysing the OPALS Feasibility Questionnaire’s answers among those participants who did not perform the intervention for more than two weeks (*n* = 14), inconsistent answers were observed. The results are summarised on Table 2.

Based on the feedback provided by the participants, one declared that she took the intervention as an opportunity to do something positive for her offspring by reducing the risks that obesity implies for her baby. Another one, decided to take part to compensate the culpability she felt for being obese. One felt somehow linked with the researcher and kept on doing the intervention not to let her down. 

## 4. Discussion

The aim of the study was to assess the feasibility of an active sitting exercise intervention designed to reduce sedentary time and increase energy expenditure, for morbidly obese pregnant women, by appraising the feasibility of recruitment, adherence, and retention. We hypothesised that using a home-based method involving simple, safe, and mostly sitting exercises would encourage at least the half of the recruited morbidly obese pregnant women to accomplish the whole proposed exercise plan. It is important to highlight that most of morbidly obese women are inactive and apparently not interested in exercising, therefore pregnancy might be an opportunity for some of them to start changing that behaviour. Thirty morbidly obese pregnant women were recruited to take part in a home-based exercise intervention. Among these, two women did not attend for final data collection. Only 21.4% (*n* = 6) completed the 12 suggested weeks of intervention, meanwhile 39.2% (*n* = 11) completed at least six weeks of intervention. This is similar to what was observed by Seneviratne et al. (2016), also using a home-based exercise strategy with overweight and obese pregnant women but achieving a low compliance [27].

Aiming to design an appropriate randomised controlled trial, our results show that the proposed exercise strategy might be feasible, as it has been shown what to expect for future studies using the same, or an improved intervention. This study has clarified the numbers that are needed to calculate a proper sample size for recruitment and for completion of an exercise intervention for morbidly obese pregnant women.

### 4.1. Study Strengths

The main strength of the exercise intervention was the inclusion of morbidly obese pregnant women in the design, which allowed us to improve and test the intervention during the design process, making it appropriate and practical. Another strength is the home-based approach, which was well received by the participants in general, and seemed suitable for morbidly obese pregnant women. Another strength of the study was the use of the time sedentary to introduce an exercise intervention. Furthermore, the fact that the intervention can be performed while sitting was also convenient, as participants were able to do other things at the same time, like watching television. The minimal and cheap materials needed to perform the exercise intervention is another strength. Finally, the simplicity of the exercises, which were chosen because did not require a complex technique, which was verified by participants who declared that the instructions were easy to follow. 

### 4.2. Study Limitations

The main weakness of the intervention was the lack of an objective measurement to study energy expenditure and time spent sedentary. Additionally, the lack of supervision was a weakness, as some women needed to be encouraged and required more support to complete the intervention. The lack of warm-up and a cool-down as part of the intervention are also acknowledged as weaknesses and should have been included. Another identified weakness was in relation to Exercise 6, which involved holding the whole weight with the arms for 20 s and was reported by participants to be the most difficult. 

Involving participants in the design of an intervention might be helpful and practical, nevertheless exercises may still need to be individually tailored to enable maximum participation. The problem with exercise 6, which meant to hold the body weight with the arms by extending the elbows whilst sitting, might have been just discomfort or wrong technique, however having options might help participants to overcome individual impairments, working the same muscles, also involving the own weight. 

### 4.3. Practice Implications

Primarily our results showed low involvement in an exercise intervention for morbidly obese pregnant women. The main reason might be the discomfort that pregnancy itself entails, which discourages women to exercise, but that discomfort is certainly worse among morbidly obese pregnant women. Furthermore, most of morbidly obese pregnant women have been inactive before pregnancy, which makes it more difficult to adopt an active behaviour. In addition, we believe that the lack of information and the lack of education on the risks associated to obesity, and the benefits of exercising during pregnancy [28], might explain that low involvement we observed in the intervention. More work is needed to understand how to teach pregnant women regarding the benefits of exercising during pregnancy. Teaching obese pregnant women regarding the risks linked to obesity during pregnancy for mothers and babies. It is also important to find better ways to involve health professionals by giving them the tools to provide realistic, effective, and attractive advice, aiming to encourage pregnant women, to be more active during gestation, including physical activity specialists involved in care, as dieticians usually are, should be a substantial input. 

## 5. Conclusions

Participants have shown a real interest in helping to design an effective exercise intervention. Involving and empowering participants in how to take care of themselves, as part of the intervention, might help to increase their commitment. Home-based approaches should fit better for morbidly obese pregnant women, facilitating their participation. Additionally, trying to avoid the effect of their body weight whilst performing the exercises should help participants to comply as it would make it enjoyable, practical, and challenging at the same time, as a reasonable load during working out means an acceptable and safe intensity.

Almost 20% of morbidly obese pregnant women who agreed to take part in our 12-week exercise intervention managed to complete it. We believe that 20% is a sufficient proportion to keep on trying interventions based on physical activity, as reducing the time spent sedentary and increasing the energy expenditure among morbidly obese pregnant women might diminish the risks associated with obesity and excessive time sedentary for those women who comply, helping them and their babies. 

Although most of the participants understood that the intervention might have been beneficial for their pregnancies, and signed the consent hoping to commit, being pregnant is complicated for most women, due to discomfort, tiredness and other feelings and emotions, but being in addition severely obese makes it even more difficult. Furthermore, some women had to keep on working, or had more children to take care of, or felt too tired, or had pain, somehow excusing them for not fulfilling the 12-week exercise intervention. From that point of view, of 28 women, 11 completed at least six weeks of the exercise programme and declared to keep on doing it, which is promising and might improve their pregnancy outcomes.

We hope this study will lead to new and better studies, ideally a randomised controlled trial which allow the effect of the exercise intervention on pregnancy outcomes to be clarified, such as the incidence of gestational diabetes mellitus (GDM), gestational hypertension, preeclampsia, and gestational weight gain (GWG), for mothers, and the risk of macrosomia for the newborn.

## Figures and Tables

**Figure 1 ijerph-18-05673-f001:**
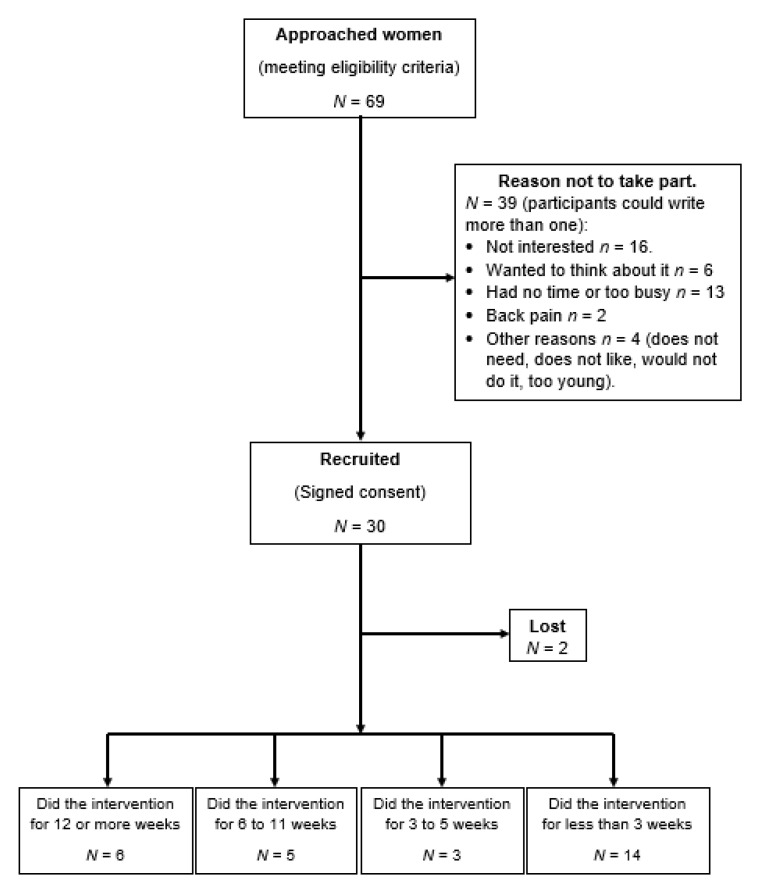
Flow chart of recruitment process.

**Figure 2 ijerph-18-05673-f002:**
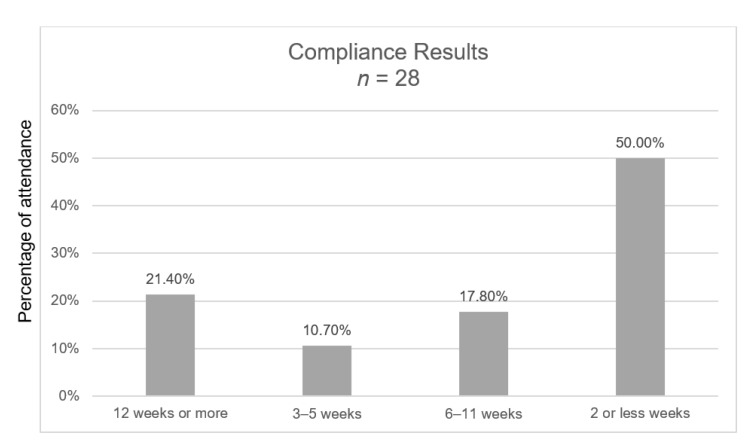
Compliance results.

**Table 1 ijerph-18-05673-t001:** OPALS feasibility questionnaire main results among participants completing six or more weeks of intervention.

Regarding the Intervention	Participated for 6 or More Weeks (*n* = 11)	Participated for 12 or More Weeks (*n* = 6)
Time performing on average	11 weeks	13 weeks
First child	5 (45.4%)	4 (66.7%)
Had children	6 (54.6%)	2 (33.3%)
Frequency per week on average	4 times per week	4 times per week
Once per day	8 (72.7%)	4 (66.7%)
More than once per day	3 (27.3%)	2 (33.3%)
Enjoyable Yes	10 (90.9%)	5 (83.3%)
Will keep on doing	9 (81.8%)	6 (100%)
Easy to participate and follow the instructions	11 (100%)	6 (100%)
Performance improved Yes	9 (81.8%)	5 (83.3%)
Made feel better	10 (90.9%)	6 (100%)
The diary was helpful Yes	10 (90.9%)	6 (100%)

**Table 2 ijerph-18-05673-t002:** OPALS feasibility questionnaire results on the reasons not to commit with the intervention, among those participants who completed the half or les of the intervention (*n* = 14).

Reasons	Frequency	Percentage
Finding the time to do the exercises	3	21.4%
Pelvic pain	3	21.4%
Too busy	2	14.3%
Sickness	2	14.3%
Different reasons each (tediousness, extreme anxiety, etc.)	3	7.2%
No answer	1	7.2%

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
