# Peer review of "Options in Pregnancy to Increase ActiveLy Sitting (OPALS) Feasibility Study"

_ijerph, 2021, doi:10.3390/ijerph18115673_

Round 1

Reviewer 1 Report

The paper presents a feasibility study of an exercise intervention for morbidly obese pregnant women to calculate essential factors, such as recruitment rate, adherence, and compliance, to inform the design of a randomized controlled trial, to test the effect of the exercise intervention on pregnancy outcomes.

Some minor issues were encountered, such as:

  • Formatting of lines 99 and 143;
  • Line 16: correct the expression “..as we as..”;
  • Line 211: correct “These is…” to “This is..”;
  • Line 212: the reference has no data;

Regardless, some methodological problems concern me such as the small sample size and the low percentage of compliance.

There is no detailed information on patients' characteristics (descriptive statistics) at baseline and at each time point.

With only 20% of compliance, I feel it is difficult to conclude that “the proposed exercise strategy is feasible” as mentioned by the authors. Although the exercise choices, the designed series are well reasoned.

Author Response

Point 1. Formatting of lines 99 and 143.

Response 1. That was corrected, and can be checked, now are lines 98 and 142.

Point 2. Line 16 correct the expression "..as we as..".

Response 2. It has been changed to "..as well as..", in line 16.

Point 3. Line 211: correct "These is..." to "This is...".

Response 3. It has been amended in line 217.

Point 4. Line 212: the reference has no data.

Response 4. Data has been added in line 218.

Point 5. There is no detailed information on patients'characteristics  (descriptive statistics) at baseline and at each time point.

Response 5. All characteristics that were collected have been reported on Table 1, and were obtained at baseline. 

Point 6. It is difficult to conclude that "the proposed exercise strategy is feasible".

Response 6. The sentence was changed for "the proposed exercise strategy might be feasible".

Reviewer 2 Report

General comments

This study assessed the feasibility of an exercise intervention on pregnant women. The study covers a very interesting topic, with recognised importance for this population. The manuscript is well structured and complete. I have only some minor comments that I hope will be useful to improve its quality.

  • I suggest to move purpose of the study from material and methods to the end of the introduction, together with an experimental hypothesis.
  • Design should be better clarified. The sentence “To assess the feasibility of an exercise intervention, a feasibility study was conducted.” seems to have no sense.
  • Please include the characteristics of participants within the text (and not as a list).
  • Line 100-105. Please consider to refer to the recent ACSM pronouncement:

“Benefits of Physical Activity during Pregnancy and Postpartum: An Umbrella Review”

  • Please improve the quality in terms of pixels definition of the Figure 1.
  • The first sentence of the Discussion should reflect what stated in the study’s aim and experimental hypothesis.

Author Response

Point 1. I suggest to move purpose of the study from material and methods to the end of the introduction. 

Response 1. Purpose has been moved, as suggested, to line 62-64.

Point 2. Add an experimental hypothesis. 

Response 2. An hypothesis was added (lines 64-66).

Point 3. The sentence "To assess the feasibility of an exercise intervention, a feasibility study was conducted" seems to have no sense.

Response 3. The sentence has been amended: "After the exercise intervention was designed and tested with morbidly obese pregnant women, we conducted the present study to assess the feasibility of the intervention". (lines 72-73). 

Point 4. Please include the characteristics of participants within the text (and not as a list).

Response 4. Now characteristics of participants are written whithin the text (lines 92-97).

Point 5. Line 100-105. Please refer to the recent ACSM pronouncement. 

Response 5. The suggested reference has been added. We appreciate and thank the suggestion, as it has been a real input not only for this paper (lines 99-104).

Point 6. Please improve the quality in terms of pixels definition of the Figure 1.

Response 6. I tried to improve the quality of Figure 1, and I believe it worked. I apologise if it is still not good enough.

Point 7. The first sentence of the Discussion should reflect what stated in the study's aim and experimental hypothesis.

Response 7. The first sentence of the Discussion has been amended acording to the suggestion (lines 206-2014).